# Digital Twins in Healthcare: Methodological Challenges and Opportunities

**DOI:** 10.3390/jpm13101522

**Published:** 2023-10-23

**Authors:** Charles Meijer, Hae-Won Uh, Said el Bouhaddani

**Affiliations:** Department Data Science & Biostatistics, Julius Center, UMC Utrecht, 3584 CX Utrecht, The Netherlandsh.w.uh@umcutrecht.nl (H.-W.U.)

**Keywords:** virtual twins, personalized medicine, precision medicine, digital twin methodology, multi-modal data sources, AI, data integration

## Abstract

One of the most promising advancements in healthcare is the application of digital twin technology, offering valuable applications in monitoring, diagnosis, and development of treatment strategies tailored to individual patients. Furthermore, digital twins could also be helpful in finding novel treatment targets and predicting the effects of drugs and other chemical substances in development. In this review article, we consider digital twins as virtual counterparts of real human patients. The primary aim of this narrative review is to give an in-depth look into the various data sources and methodologies that contribute to the construction of digital twins across several healthcare domains. Each data source, including blood glucose levels, heart MRI and CT scans, cardiac electrophysiology, written reports, and multi-omics data, comes with different challenges regarding standardization, integration, and interpretation. We showcase how various datasets and methods are used to overcome these obstacles and generate a digital twin. While digital twin technology has seen significant progress, there are still hurdles in the way to achieving a fully comprehensive patient digital twin. Developments in non-invasive and high-throughput data collection, as well as advancements in modeling and computational power will be crucial to improve digital twin systems. We discuss a few critical developments in light of the current state of digital twin technology. Despite challenges, digital twin research holds great promise for personalized patient care and has the potential to shape the future of healthcare innovation.

## 1. Introduction

Digital twin technologies have seen a rise in popularity in various industries, including manufacturing, engineering, and rocketry, from where the term originated [1]. This rise can be attributed to the developments in rapidly collecting, storing, and sharing data, together with computers being able to apply complex models and algorithms in a short amount of time [2]. In several fields of healthcare, such as precision medicine, clinical trials, and public health, the application of digital twins has become more and more apparent as they may serve as a tool to understand and simulate complex physiological processes. Moreover, digital twins may also reduce the need for animal experimentation, which takes an estimated 200 million animals per year [3], as it allows a direct translation of in vitro measurements into what could be expected in vivo either in digital animal models or humans [4].

General definitions of a digital twin have been given in the literature [5,6,7,8,9]. In this narrative review, we work with the general definition of healthcare digital twins [10] as virtual replicas of real human patients, through which clinicians can gain valuable insights, optimize treatment strategies, and deliver personalized care [5,11,12]. For specific healthcare domains, the operationalization of this definition depends very much on the underlying methodology and data used to construct the digital twin. Though the general aim is expected to align with the definition above, a ‘*cardiac*’ digital twin (e.g., Section 2.2 below) differs considerably from a ‘*drug response*’ digital twin (e.g., Section 2.6 below) in methodology, data types, and implementation. One of our goals is to take a deeper look into the methodological aspects underlying digital twins across several healthcare domains where this technology is being applied. By gaining an understanding of the methods and data used, the potential value and important pitfalls of digital twins in future studies can be more easily identified.

Healthcare digital twins require large amounts of data and often multiple data types. These include measurements that can be made using a smartphone or -watch, like heart rate, temperature, and location [13], and data that can otherwise be gathered at home, such as blood pressure and blood oxygen saturation, but also medical imaging data recorded during CT or MRI scans, electrophysiology, and various types of -omics data, which can be collected through a wide range of techniques including sequencing, immunoprecipitation and mass-spectrometry [14]. After generating the digital twin, a variety of methods can be applied to make simulations and predictions. These range from fitting regression lines to the data to designing deep neural networks [15] and can be used for many different purposes. Apart from their clinical use, digital twin technologies may also be applied to identify novel drug targets, simulate the effectiveness and safety of new treatments, or predict patient traffic during a pandemic [5].

The aim of this narrative review is twofold. We aim to review the methodological development of digital twin systems across several healthcare domains. Second, we aim to identify the types of data required to construct the respective digital twins. Secondary to these aims, we further discuss how to overcome the challenges introduced by handling large amounts of data and standardization, integration, and interpretation of many different types of data. The research questions we desire to answer are: (1) which data types and sources are important for the development of healthcare digital twins? (2) What are the prevailing methods and techniques employed in healthcare digital twin systems, and how do they vary in their applications? (3) How can the challenges related to healthcare digital twin methods and data be transformed into opportunities? Addressing these aims and questions will be crucial to harnessing the full potential of digital twins, ensuring that this promising idea can be integrated successfully into clinical practice.

## 2. Case Studies

Digital twins of complex systems require vast amounts of data to accurately represent their physical counterparts. Different types of data can be gathered via different methods, and integrated into one model, to simulate pathways, organs, or entire organisms. However, gathering all that data may be especially challenging in medical care, compared to the original DT application in rocketry, for example. While a lot of environmental data can be continuously captured using body-worn sensors like a smartphone or -watch, more complicated and intrusive methods may be necessary to gain -omics or imaging information [5]. We present several key case studies in different healthcare fields where we review the data sources used and the methodology applied to construct digital twins. To aid in quickly searching the relevant literature regarding digital twin methodology, an annotated overview of the significant literature is presented in a searchable spreadsheet as Appendix A. Together with this supplementary overview, the case studies reviewed below provide an overview of the methodological cornerstones of digital twins across different healthcare domains.

### 2.1. Artificial Pancreas

One of the first digital twin-like systems is the artificial pancreas. It consists of two essential parts: a system capable of continuously measuring blood glucose levels, and a device containing a syringe used for insulin infusion when needed. Blood glucose levels used to be measured by having the user collect a drop of blood from their finger, but less invasive methods have been developed during the late 20th century and beyond. Instead of measuring true blood glucose levels, these values could be inferred by monitoring the glucose concentration in extracellular space. However, as the relation between the new interstitial glucose data and blood glucose levels is not one-to-one, the devices had to be calibrated with true blood glucose readings. Furthermore, even after successful calibration, this method was prone to loss of sensitivity and random noise. Addressing these issues is essential in order to be usable as a ’near-future’ digital twin: for example, if glucose levels are predicted to be too high or too low in the near future, the system can generate preventive alerts, prompting the patient to take appropriate actions, such as adjusting insulin dosage or dietary choices.

To combat these issues, a collection of signal processing algorithms has been applied to ensure accurate prediction of blood glucose levels based on minimally invasive, interstitial readings [16]. The ‘smart continuous glucose monitoring sensor’ combines the existing glucose monitoring sensor with several software modules designed to reduce noise, improve accuracy, and predict future glucose concentration (Figure 1) [17].

Denoising is used to improve the subcutaneous glucose concentration readings from the sensor by reducing the impact of noise in the data. To estimate the true interstitial blood glucose concentration, the denoising algorithm uses a Bayesian interference algorithm that takes into account general information on signal-to-noise ratios, as well as the data it has collected previously from the specific individual, to determine which parts of the signal are noise. The algorithm also does not require user intervention and is designed to be adaptive to the signal-to-noise ratio of every individual user. Further, to combat under- and overestimations of blood glucose levels based on subcutaneous glucose readings, the data is enhanced using a least squares linear regression model. Briefly put, blood glucose measurements are fit against blood glucose estimations made based on the interstitial readings taken at the same time. Then, the regression parameters are used to enhance future data collected by the subcutaneous sensor to more accurately estimate the corresponding blood glucose values. This linear regression can be updated in real-time, and it takes into account the influence of blood-to-interstitium glucose transport and its delay on the individual user. The addition of these two data processing steps resulted in a greatly improved blood glucose estimation accuracy, which is essential for devices using subcutaneous readings (Figure 2). Lastly, the smart sensor is capable of predicting future glucose concentrations to enable the device to generate timelier alerts. Glucose level prediction is achieved by reading all the past data generated by the sensor and assigning every measurement a different weight, based on an autoregressive model. The future values are subsequently calculated by multiplying each past data point by its weight in real-time, and a preventive alert can be generated when the predicted value is either too low or too high [17].

In the last few years, glucose monitors using these algorithms have been approved for use by the FDA without the need for calibration by capillary blood readings. These devices are capable of measuring patient data, applying data processing, and predicting future values in real-time [16]. However, many more variables other than blood glucose are needed to create a complex pancreas digital twin. In 1979, the rate of glucose processing was described in a nonlinear function [18] and that model has evolved into one describing a glucose–insulin network using many functions and parameters to take into account the glucose kinetics, insulin kinetics, rate of glucose appearance, endogenous glucose production, utilization, secretion and excretion [19]. Furthermore, models that describe the effect of external influences like physical activity and the delays associated with subcutaneous, rather than intravenous, insulin delivery were designed. These models make it possible to test the effect of any meal or insulin injection, as well as any extreme scenarios digitally, before use in clinical trials. Currently, an increasing number of variables are being added to the artificial pancreas systems. Heart rate monitoring, motion sensing, additional hormones, and glucagon have all been analyzed for their use in mitigating hypoglycemia during physical exercise. Technical developments, like the prevalence of smartphones capable of running algorithms and a wireless connection, may offer patients better monitoring of their glucose levels using a device that is already integrated into daily life as the controller [16].

### 2.2. Cardiac Digital Twins

In healthcare, generating digital twins to mimic a human organ has seen much popularity in cardiovascular research. Multiple types of data are combined to create a cardiac digital twin (CDT), which can be used to test patient-specific monitoring and treatment strategies (Figure 3). The process of creating a CDT can be split into two distinct stages; the anatomical and functional twinning stages. The anatomical twinning stage consists of creating a very detailed 3D copy of the physical twin, based on CT or MRI scans of the patient [20]. This cardiac 3D mesh is based on Universal Ventricular Coordinates. This model essentially describes the location of specific cardiac regions such as the apex or septum and can be automatically computed with relatively little input data. It requires an epicardial apex surface point, a left ventricular endocardial surface point, a right ventricular septal surface point, and surface points of the ventricular base, to compute the ventricular coordinates of the heart base, epicardium, left ventricular endocardium, and right ventricular endocardium and septum. The UVC algorithm is also capable of computing coordinates for trabeculae and certain heart valve openings [21]. A similar approach can be used to map Universal Torso Coordinates.

The input data is recorded during the MRI study of the patient. The 3D whole-heart MRI scans are segmented automatically by a convolutional neural network, and corrected manually. Automatic UVC computations are then run to create the cardiac mesh for the specific heart [20].

The second stage in creating the CDT, functional twinning, covers the electrophysiology of the heart. Four factors responsible for the ECG waveforms during activation and repolarization were defined mathematically: depolarization caused by the His–Purkinje system and distribution to the subendocardium, the conduction velocity within the ventricles, spatially varying action potential duration, and the conductivity of the torso surrounding the heart. Electrophysical activity of the anatomical reference frame was simulated using a fast-forward ECG model, and compared to clinical measurements of 12-lead ECGs (Figure 4). These comparisons show that with this two-stage twinning method, cardiac electrophysiology can be simulated automatically and in near real-time.

### 2.3. Single-Cell Flux Analysis

Apart from organ-specific measurements like heart electrophysiology and blood glucose levels, the vast amounts of data generated in omics research may also be used in generating digital twins. In cancer research, single-cell digital twins based on metabolomics and fluxomics, the analysis of production and consumption of metabolites, have been proposed as a tool to better discriminate between cancer phenotypes. The model used to create the single-cell digital twin integrates single-cell RNA (scRNA) sequencing data and extracellular metabolite fluxes to obtain a view of the single-cell metabolic phenotype at any given time [22].

The single-cell Flux Balance Analysis (scFBA) model requires three types of input (Figure 5). Firstly, it needs a template metabolic network, describing the different metabolites, their biochemical reactions, and their consumption or secretion [22]. The complete human metabolic network has been reconstructed by integrating pharmacogenomic associations, large-scale phenotypic data, and structural data for proteins and metabolites. The metabolic fluxes in this network have been predicted by models that have also been fed data from other omics analyses, describing the pathways that are expressed in any given cell or tissue [23]. Secondly, the scFBA model is given an scRNA-seq dataset that contains the normalized read count of each gene in each cell in the analysis. Lastly, extracellular fluxes in the patients’ cell population are approximated from the measurement of metabolite concentrations in the cell culture medium of the patient-derived organoid or xenograft, for example.

The scFBA pipeline starts with pre-processing, by removing genes with zero read count from the template metabolic network. Then, a population model is generated. This model is created by integrating all the RNA data of all the available cells in the template metabolite network. The resulting network corresponds to the scRNA of the average cell in the sample and is copied to produce a population model consisting of replicas of the single metabolic network. All the cells in the population now have the same set of metabolites as the template network. Each single-cell network can be reconstructed by introducing cooperation reactions, which allow metabolite exchange among cells and with the environment. These reactions are then linked to the scRNA data via logistically expressed rules. The ‘AND’ operator is used to describe genes that encode for different subunits of the same enzyme, while the ‘OR’ operator describes genes that encode for isoforms of the same enzyme. These logical operators are then used to calculate the reaction activity scores for each reaction. These scores represent the expression of the genes in transcripts per kilobase million. For the reactions that are caused by genes that are linked only through the AND operator, all the genes are necessary. This means that the reaction activity score can be defined as the expression of the least-expressed gene that is necessary for that reaction to occur. If the genes involved in the reaction are only linked through the OR operator, the activity score is calculated as the sum of the expression values.

After the population model and the reaction activity scores for each cell are computed, bulk and single-cell constraints are imposed. These represent boundaries on the metabolite exchanges with the environment and within the cells based on their reaction activity scores, respectively. The model, now describing the metabolite exchange between single cells and the environment, constrained by the reaction activity scores for each cell, as well as by boundaries set through measurements of the entire sample, can be used to simulate the effect of single gene deletions. The reactions that are associated with that gene which is only linked to other genes by the AND operator should be disabled by the deletion. These reactions are removed from the network, and the population model is reoptimized for total biomass production. This allows analysis of the effect of single genes on tumor growth in a patient-specific cell system, which may lead to identifying genes or clusters of cells that can be exploited to deregulate cancer metabolism [22].

### 2.4. Protein and DNA Interactions

Networks like the ones generated with scFBA may be created and applied in digital twin computations for other -omics data, as well. Protein interactions may be studied through multiple techniques. The yeast two-hybrid and LUMIER methods can both be applied to check for interaction between two specific proteins [24,25]. A high-throughput platform combining immunoprecipitation and high-throughput mass spectrometry (IP-HTMS) is capable of rapidly identifying novel protein interactions for a protein of interest (Figure 6).

To demonstrate the IP-HTMS workflow, 407 ‘bait’ proteins of interest were flag-tagged and isolated, together with any interacting ‘prey’ partners, via immunoprecipitation. The proteins were then subjected to SDS-PAGE and mass spectrometry for identification. All proteins and peptides that were associated with the same bait protein were clustered and an ‘anchor’ protein was selected for each cluster by ranking the proteins within the cluster based on the number of peptides. Interactions that were non-specific, bait–bait interactions, and interactions with contaminant proteins were removed from the interaction network. Several metrics were used to generate a measure of confidence in the bait–prey interactions, and high-scoring pairs were further analyzed by integrating other types of genomic information, such as gene expression, sub-cellular location, and function. With this pipeline, many protein interactions may be studied rapidly to create complex protein–protein interaction networks [26]. Networks like these can provide critical information to human digital twins, as they enable in-depth analysis of the effects of the absence or abundance of specific proteins on their pathways, which can lead to understanding why certain diseases occur, as well as pinpointing potential targets for treatment.

Protein–DNA interactions can also be analyzed, although the regulatory networks are more incomplete in comparison with protein–protein interaction networks, metabolic networks, and RNA networks [27]. Chromatin immunoprecipitation (ChIP), combined with next-generation sequencing can be used to identify DNA-bound proteins, as well as the DNA sequence they are bound to. This information may explain the effect of an altered DNA sequence if it results in a transcription factor not being able to interact with the DNA, for example. Protein–DNA interactions uncovered via ChIP-sequencing have been reported in databases like UniPROBE and JASPAR [28,29], but the technique is limited by the cost as well as the availability of the high-quality antibodies needed to retrieve the DNA–protein complexes [27].

### 2.5. Clinical Reports in Oncology

Advancements in machine learning and specifically natural language processing (NLP) have enabled the use of written records in creating digital twins. In cancer research, structured, written reports containing ‘findings’ and ‘impressions’ from CT-scan analysis of multiple organs were annotated for the presence or absence of metastases by five radiologists (Figure 7). Individual reports from each patient were concatenated in chronological order to enable multi-report analysis. This allows the model to access every previous report when it predicts the presence of metastases during the time of a patient’s third report, for example. This is especially important in the event of no change compared to the last analysis being reported for a particular organ. The multi-report analysis enables the algorithm to decide whether ‘no change’ means an analysis based on the previous information.

The structured reports are first converted into numeric vector representations to be used as inputs for the three machine learning models developed to predict metastasis presence over time. This can be achieved by removing the punctuation and unknown words in the report and assigning each word an index value. The strings of index values representing the written text are fed as input to the convolutional neural network (CNN), capable of learning which combinations of words mean the presence and which mean the absence of metastasis in each analyzed organ. An augmented CNN with an attention layer is used to better capture important information in the reports by assigning higher weights to the indices that represent more important words. Thirdly, to take the context into account, a bi-directional long short-term memory (LSTM) network was developed. This variant of a recurrent neural network is capable of processing the data both forward and backward. This allows the network model to account for both past and future contexts when learning the meaning of different word combinations. The LSTM network is designed to deal with long sequences, and it can determine what information needs to be remembered and what can be forgotten [30]. The three models were tested on over fourteen thousand radiology reports on the lung, liver, and adrenal glands. Prediction accuracies exceeded 96% across all combinations of models and organs. This shows that with the use of NLP algorithms, written report data could contribute to developing a cancer digital twin, as these texts still contain much of the information in the medical record [30].

### 2.6. Predicting Drug Effectiveness

Once available, healthcare digital twins may be used to predict treatment outcomes, simulate various events, or digitally test the effects of the absence of a certain protein, for example. Different goals require different statistical methods and, just like during the creation of the twins, speed, and accuracy are key in creating viable digital twin applications.

During clinical trials of a new treatment, the efficacy of the new treatment is usually tested against a standard treatment or a placebo when given to a random sample of the population. However, the new treatment could only be more beneficial to a select subgroup of patients in the sample. It is worth trying to analyze what characteristics define this subgroup, to understand why the treatment is especially effective for them [31]. Although selecting a couple of features to create subgroups is known to be prone to finding false positives, various statistical methods have shown to be capable of this task [32].

The classical method consists of fitting a regression model based on the interactions between treatments and patient data. One drawback of this model is that it is not suitable for use with datasets containing many different variables, as it would need to consider many different possible interactions [33]. Multiple new algorithms for defining the boundaries of a certain subgroup have been tested.

One method relies on the use of random forests and regression or classification trees to prioritize covariates that predict which patients will benefit most from a treatment. First, random forests are applied to the data which take the variable values, including the treatment group, as input, and give the probability of a certain outcome as output. The estimated treatment effect is subsequently calculated by subtracting the probability of a positive outcome under control from the positive outcome after treatment. So, a high estimated treatment-effect value means that the treatment greatly affected the patients’ chances of a positive outcome. Then, the variables that have a strong effect on the estimated treatment-effect value are selected through either regression or classification trees. With the regression tree method, a regression tree is created with the estimated treatment effect as the response and the variables as the other input data. This tree is used to again predict the treatment-effect value for each patient, and patients with a high estimated value are grouped. The variable values that result in an increased effect can be found by analyzing the tree and finding the paths that lead to terminal nodes with high predicted-effect values [31]. With the classification method, the estimated treatment-effect value is dichotomized by splitting the outcomes using a threshold value. This binary estimated treatment-effect value is used to generate the classification tree that is used to classify the patients into either the ‘low effect’ or ‘high effect’ groups. This means that every variable used by the tree to classify a patient in the ‘high effect’ group can also be used to define a digital twin [31].

The random forest and regression tree approaches, as well as the classical model, have been tested on data from a clinical trial in 1019 patients, 517 of whom received the experimental treatment, while the others received a placebo. The patient’s condition was possibly fatal, so the positive outcome was defined as survival 28 days after receiving the treatment or placebo. Both the regression and classification methods resulted in the identification of variables that could be used to define the subgroup of patients to whom the experimental treatment was especially beneficial. The models identified four variables that affected the estimated treatment effect the most, three of which were related to the severity of the patient’s condition [31]. In this case, the differences in treatment effectiveness between patients in the subgroup and the average patient were not convincing enough to definitively prove that patients in the subgroup have a significantly better outcome. However, it shows that these digital twin-centric methods are an improvement on the classic logistic regression method when it comes to identifying and defining a subgroup of patients during clinical trials. The methods are more suited to bigger datasets, easier to interpret, and better at defining subgroup boundaries. Developments like these are essential for the application of digital twins in healthcare research.

### 2.7. Drug Repurposing for SARS-CoV-2

When the SARS-CoV-2 (or COVID-19) pandemic began, a lot of research was carried out to find agents that could either prevent or cure a COVID infection in a relatively short time. One quick way of obtaining suitable drugs on the market was to find drugs that had already been approved for use in another healthcare application and repurpose them for COVID treatment. One study started by searching for drugs that were approved for diseases with a similar molecular effect as COVID-19 [34]. To find these drugs, 332 host protein targets of the coronavirus were mapped to the human interactome. Of these targets, 208 turned out to be connected within the interactome network.

Three methods were used to identify potentially repurposable drugs for COVID-19 treatment (Figure 8). Firstly, an AI-based algorithm was used to map drug–protein targets and disease–protein targets. Secondly, a diffusion algorithm ranked the available drugs based on their ability to affect the pathways that contained the SARS-CoV-2 protein targets in the network. Lastly, a proximity algorithm was applied to calculate the distance between the host protein targets of SARS-CoV-2 and the closest proteins that were targeted by the drugs.

The predictions made using the three methods were compared to compounds that had been experimentally screened for their efficacy in SARS-CoV-2 in monkey kidney cells. Of the 918 tested drugs, 77 had a positive effect, 806 showed no effect, and 35 turned out to be toxic to the cells. The drugs were subsequently compared to another dataset containing outcomes of clinical trials, as well [34]. Lastly, the drugs were given a rank based on their scores in the different pipelines described above. Multiple rank aggregation algorithms were tested for this purpose. CRank, capable of extracting the predictive power of the individual methods, consistently showed a strong predictive performance among datasets [34]. Of the 200 drugs ranked by this algorithm, 13 had positive outcomes in the monkey cells. Two drugs were already tested repeatedly, and of the remaining eleven, six showed potential for treating SARS-CoV-2 infection when tested on human cells. Of these drugs, three were highly ranked by CRank and had strong outcomes in the experimental tests, but were not yet used in clinical trials [34].

Studies like these show how digital twins containing protein–protein interaction networks may be used to estimate the effect of new treatments. In this case, a general interactome network was used to find drugs that may be potent in the treatment of COVID-19 infection in a population. However, these methods may also prove useful for a personal digital twin-based approach to evaluate treatment options for specific individuals.

## 3. Discussion

Recent developments in digital twin research in healthcare show great promise in understanding complex physiological processes, and may be applicable in several medical fields. To fully grasp the possibilities and pitfalls of these digital twins, we provide an in-depth look into the methodology and data types used in constructing digital twins. Though unified by their common goal, digital twins from different fields vary considerably in methods and data used. The main findings are summarized in Table 1.

There exist several opportunities and challenges in the methodology and data types underlying digital twins. In diabetes management, the artificial pancreas is a prime example of a digital twin-like system that can greatly improve the ability to monitor and predict blood glucose levels and administer insulin based on non-invasive glucose monitoring methods. The data and methodology employed in the ‘smart sensor’ represent a multifaceted and data-driven approach, namely continuous glucose monitoring, signal processing algorithms, Bayesian interference, least squares linear regression, and autoregressive models. While this ‘smart sensor’ system can already offer great benefits to diabetes patients, there is still much room for improvement. The denoising and data enhancement methods could be combined and performed at the same time to reduce the complexity of the pipeline. Furthermore, the prediction module could be expanded to account for information such as meals, sleep, or physical activity. Even with these challenges that still need to be addressed, the reliability and effectiveness of these systems are demonstrated by the fact that the FDA has already approved them for personal use in their current state.

In cardiovascular research, a field that has seen many new ideas and improvements in digital twin systems recently, multiple types of data, including MRI and CT scans and electrophysiology measurements, can be integrated to compile a digital heart model through which ECG patterns can be simulated and predicted in any location and in real-time. This ability allows for the testing of patient-specific monitoring and treatment strategies and has the potential to significantly improve patient outcomes in cardiovascular diseases. The approach here consists of two distinct stages: anatomical twinning and functional twinning. In the anatomical twinning stage, detailed 3D representations of the heart are generated based on patient-specific CT or MRI scans. The second stage focuses on the electrophysiology of the heart, where mathematical models are used to describe and simulate electrophysiological activity of the heart. One important limitation of this method is that it requires accurate, multi-label segmentation to create anatomical CDTs. This is the largest computational time sink in the whole pipeline. While some studies have shown neural networks trained for this purpose, a fully automated method for segmentation of the cardiac chambers is not available at this time. The same is true for the segmentation of the torso. Models capable of fully automatically threshold-based segmentation that account for patient-specific anatomy will have to be achieved in the future. Additionally, the representation of the His–Purkinje System in the CDT may be too simplistic, as its workings are not yet fully understood. However, it was noted that once an activation profile has been identified, an automated workflow for integrating a topological representation of the HPS can be readily implemented [20].

Ultimately, a fully comprehensive healthcare digital twin would also require the integration of different types of omics data. The scFBA model can be used to incorporate single-cell RNA sequencing data and extracellular metabolite fluxes into digital twins that can provide insights into the metabolic phenotypes of cancer cells and allow for the analysis of the effect of genetic alterations. In the scFBA model, three main types of input are required: a template metabolic network, a single-cell RNA sequencing dataset, and approximated extracellular fluxes from the patient’s cell population. Using these data, gene-to-metabolic reaction links are calculated and used to simulate the effect of single gene deletions. This approach offers a powerful tool to study cancer phenotypes and identify potential novel targets for therapeutic intervention.

Furthermore, the developments in machine learning and AI-based statistical methods allow for the prediction of drug effectiveness in patients (regression and random forests), identification of already approved drugs that may be repurposed for another cause (drug- and disease-protein mappings and rank aggregations), and the ability to extract data from clinical reports (natural language processing and neural networks), for example.

However, several challenges need to be addressed. Firstly, while the metabolic network maps are very comprehensive, protein–protein and regulatory networks are still considered incomplete. Gathering and integrating large amounts of data from diverse sources remains a significant hurdle and the continuous development of non-invasive and high-throughput data collection methods will be crucial to improve the accuracy and effectiveness of digital twin-based approaches. This is especially important for strategies that rely on digital twin systems to monitor health in real-time, to be able to predict a drop in blood glucose levels or to generate alerts based on simulated EEG patterns, for example. Additionally, a wide range of variables and parameters in digital twin models is necessary to accurately mimic complex physiological systems. Integration of these variables requires ongoing advancements in computational power and modeling techniques. Lastly, it is paramount to the practical application of digital twin systems that the privacy of the patient can be guaranteed, while large amounts of data are collected and ideally shared, to enable researchers to collaborate all over the world.

A lot of research is already being carried out to overcome these obstacles, and access to an ever-increasing amount of computational power allows for the use of more and more data, as well as complex models and algorithms. An increasing number of publicly available template models describing protein interactions, single-cell metabolomics, and thoracic cell coordinates, for example, will also be paramount in creating patient-specific digital twin systems in a short time. Additionally, modern systems and standards for data management will enable secure and efficient ways to store personal data and share them with others all over the world.

In conclusion, the development of digital twins in healthcare has the potential to revolutionize medical care and personalized treatments. The examples discussed in this review demonstrate the effectiveness of digital twin technology in artificial pancreas systems, cardiac digital twins, and single-cell digital twins for cancer research. By integrating diverse data sources and advanced modeling techniques, digital twins offer a powerful tool to simulate and understand complex physiological processes. Continued research and development in this field will pave the way for improved patient care and precision medicine.

## 4. Conclusions

Recent developments in digital twin research in healthcare hold great promise for understanding complex physiological processes and their potential applications in various medical fields. The examples discussed include digital twins in diabetes management, such as the artificial pancreas, which improves blood glucose monitoring and insulin administration. Challenges exist in denoising data and expanding prediction modules. In cardiovascular research, digital heart models enable real-time prediction of ECG patterns, allowing for patient-specific monitoring and treatment strategies, yet segmentation and representation challenges persist. Integrating omics data and AI-based methods in comprehensive healthcare digital twins provides insights into cancer phenotypes and drug effectiveness prediction. Addressing challenges in data integration, computational power, and privacy, is crucial for advancing digital twin-based approaches. Overall, digital twins have the potential to revolutionize medical care and precision medicine, offering personalized solutions for patients.

## Figures and Tables

**Figure 1 jpm-13-01522-f001:**
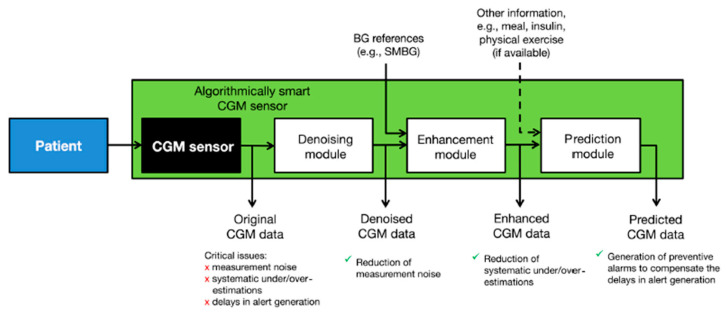
Schematic of the smart continuous glucose monitoring sensor which allows for subcutaneous glucose reading, signal processing, and future reading prediction to reduce measurement noise and under- and overestimations of blood glucose values. It also contains a prediction module to generate timelier alerts. Figure taken from [17].

**Figure 2 jpm-13-01522-f002:**
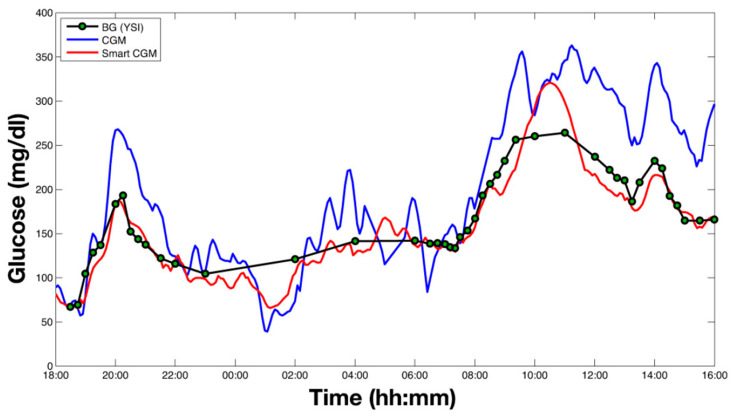
Results of application of the smart continuous glucose measuring system on a human subject. Original reading data are in blue, denoised and enhanced values are in red, and reference blood glucose measurements are shown as green dots. Figure taken from [17].

**Figure 3 jpm-13-01522-f003:**
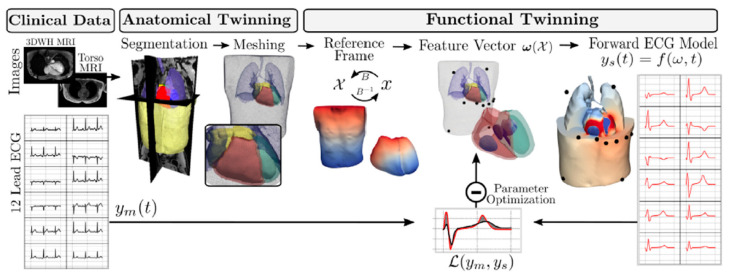
Schematic of workflow for CDT generation. MRI data is segmented and used to create anatomical meshes. A reference frame (X) is computed based on UVC and UTC. ECG waveforms generated with a forward ECG model are compared to clinically measured 12-lead ECG data for optimization of the model parameters contained in w(X). Figure taken from [20].

**Figure 4 jpm-13-01522-f004:**
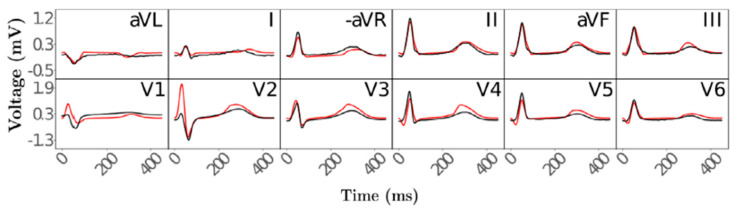
Simulated ECGs (red) compared to measured reference ECGs (black) after obtaining optimal parameter values. Electrodes were placed and simulated according to the 12-lead ECG setup. Figure taken from [20].

**Figure 5 jpm-13-01522-f005:**
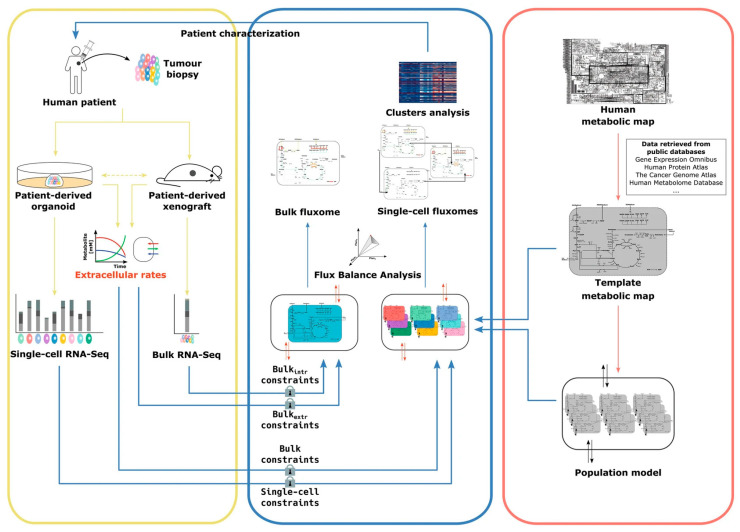
Schematic overview of the scFBA workflow. Single-cell RNA-seq and bulk RNA-seq are performed on patient-derived organoids or xenografts. Extracellular metabolite exchange rates are also measured. A template metabolic network is imported from a public database. The bulk RNA and flux data are integrated to form a population model. Single-cell networks can be computed by incorporating bulk constraints based on bulk data, and single-cell constraints based on the single-cell RNA-seq data. Figure taken from [22].

**Figure 6 jpm-13-01522-f006:**
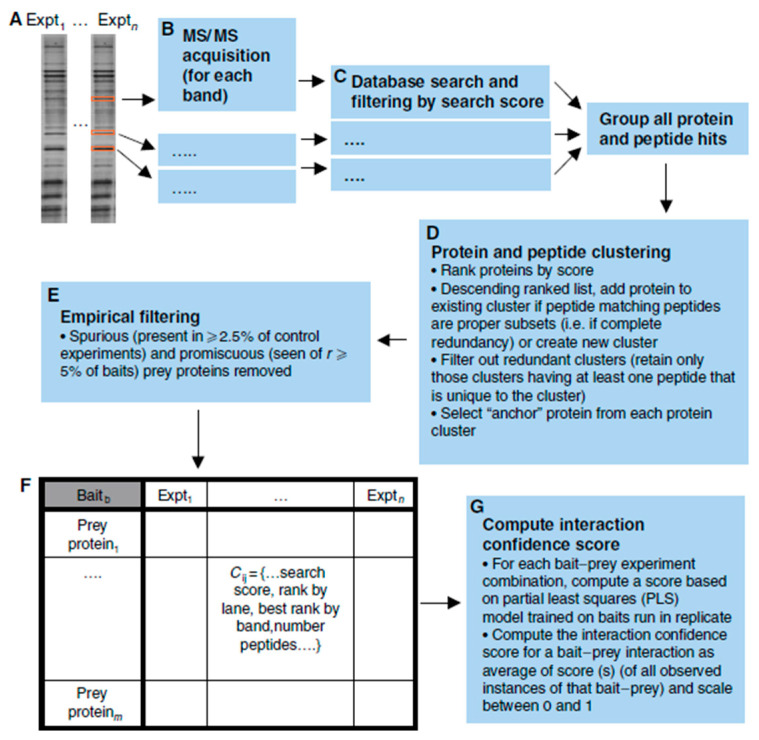
Schematic of the IP-HTMS pipeline. Bait proteins were isolated with their prey partners through immunoprecipitation and identified after SDS-PAGE and mass spectrometry. The proteins and peptides are clustered based on a scoring system and filtered. Finally, a confidence score is calculated for each bait–prey interaction. Figure taken from [26].

**Figure 7 jpm-13-01522-f007:**
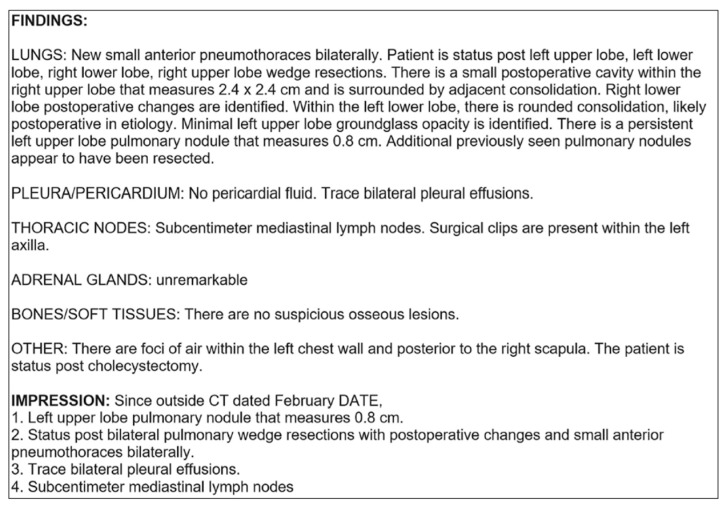
Example report of a chest CT containing organ site-specific ‘findings’, and ‘impressions’, with information about any organ. Figure taken from [30].

**Figure 8 jpm-13-01522-f008:**
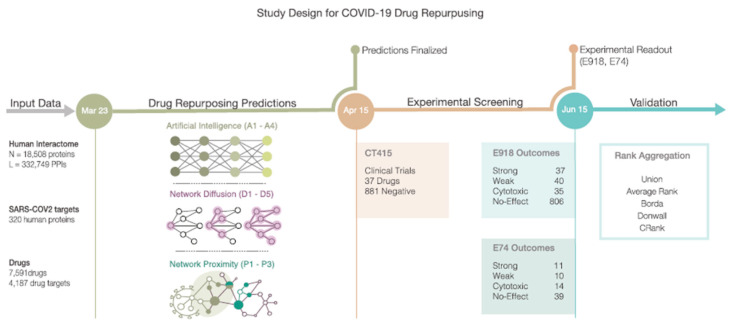
Schematic of the workflow for identifying drugs that could be repurposed for COVID-19 treatment. An AI-based method, network diffusion, and network proximity were used to identify possibly repurposable drugs, which were then validated through comparison with various clinical trials. Various rank aggregation algorithms were used to rank the drugs, based on their results in the different pipelines. Figure taken from [34].

**Table 1 jpm-13-01522-t001:** Summarizing overview of the various methodologies and data types used to construct digital twins across several healthcare fields.

Case Study	Aim of Digital Twin	Input Data	Methodology
Artificial Pancreas	Enhance blood glucose level monitoring and insulin delivery for individuals with diabetes, ensuring accurate predictions, noise reduction, and timely alerts without the need for frequent calibration.	Blood glucose data collected non-invasively via continuous monitoring. Calibration data for accurate glucose level predictions. Data related to glucose–insulin networks, external factors (e.g., physical activity), and additional hormones.	Signal processing algorithms for denoising and data enhancement. Bayesian inference for denoising. Least squares linear regression for data enhancement. Autoregressive modeling for future glucose concentration prediction.
Cardiac Digital Twin	Create detailed replicas of the heart (anatomical twinning) and simulate cardiac electrophysiology (functional twinning) for personalized testing and treatment strategies.	Three-dimensional heart scans from MRI. Clinical ECG measurements.	Universal Ventricular Coordinates for anatomical twinning. Mathematical models for cardiac electrophysiology. Fast-forward ECG modeling. Near real-time simulation of cardiac electrophysiology.
Single-cell Flux analysis	Integrate single-cell RNA sequencing data and metabolite fluxes to understand single-cell metabolic phenotypes, particularly in cancer research, aiding in phenotype discrimination.	Template metabolic networks. scRNA-seq datasets. Extracellular flux measurements.	scFBA model for metabolic analysis. Logical operators to calculate reaction activity scores. Constraints for metabolite exchanges.
Protein and DNA interactions	Construct protein–protein interaction networks for studying protein interactions and regulatory networks for protein–DNA interactions, enabling a deeper understanding of various biological processes and disease mechanisms.	Protein interaction data from techniques like IP-HTMS. Protein–DNA interaction data from ChIP-sequencing.	Bioinformatic analysis to identify and prioritize interactions. Integration with other genomic information for comprehensive analysis.
Clinical reports in oncology	Utilize natural language processing (NLP) to extract valuable information from clinical reports, particularly in cancer diagnosis, enabling better analysis and prediction of metastases presence over time.	Structured clinical reports from CT scans. Concatenated reports for multi-report analysis.	NLP for text processing. Machine learning models, including CNN and LSTM, for prediction. Multi-report analysis to improve accuracy.
Drug effectiveness	Identify subgroups of patients who may benefit from specific treatments during clinical trials, providing a more personalized and efficient approach to treatment evaluation.	Patient data and treatment outcomes. Variables describing patient characteristics.	Random forests, regression trees, and classification trees. Identification of variables affecting treatment effectiveness. Subgroup definition based on variables.
Drug repurposing for SARS-Cov-2	Identify existing drugs that can be repurposed for COVID-19 treatment by analyzing their interactions with the virus’s protein targets and predicting their efficacy, thereby accelerating drug discovery for the pandemic.	A total of 332 host protein targets mapped to the human interactome. Experimental and clinical trial outcomes.	AI-based algorithms for drug mapping. Diffusion algorithms for pathway analysis. Proximity algorithms for target prediction. Rank aggregation for drug prioritization.

## Data Availability

Not applicable.

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
