# Peer review of "Digital Twins in Healthcare: Methodological Challenges and Opportunities"

_jpm, 2023, doi:10.3390/jpm13101522_

Round 1
Reviewer 1 Report
Review Comments:
This review article mainly showcased a few case studies in constructing and applying digital twins in different areas of the healthcare industry, with the idea of identifying the available data sources and methodologies that can be targeted to various patient groups. The manuscript is in alignment with the scope of the Journal of Personalized Medicine, and it may be of interest to the readers. However, some critical points need to be addressed before publication. The reviewer recommends a major revision, with specific comments below.
1. Abstract Line 8 and related descriptions: Here the authors define digital twins to replicate patients. This “definition” or definitive statement can only be applied to this specific manuscript. Instead of confusing readers with this broad statement, the authors should revise and offer the definition for their specific case.
2. Introduction Line 31: Again, “healthcare” alone is a very general term, and it can have different meaning to readers. The authors can better define the overall scope of the term and the application area.
3. Introduction paragraph starting line 51 and the leading paragraph of Section 2: This reviewer is unsure if the case studies presented are “reviews” of other papers or specific demonstrations developed by the authors. This distinction is important, and the authors should make it clear early in the paper.
4. Section 2 Line 67-68: What is the relevance between the case studies and the annotated overviews of papers in the supplemental material? Why is this supplemental material being mentioned here? Please provide further explanation and connections in the text.
5. Before going into specific case studies, the authors are recommended to provide their understanding or definition of “digital twin”, applying specifically to this application area. In the current manuscript, such an understanding/scope does not seem to be clear. More importantly, each case seems to have a different meaning, making it hard to grasp an overall structure. If the authors intend to highlight different components in constructing a digital twin, then both the overall framework and the component should be highlighted, preferably in figures.
6. Supplemental material: The authors have highlighted this summary Excel file in a couple of instances, but the overall idea is unclear to the reviewer. It seems to cover digital twin topics in specific healthcare areas but omits important papers on digital twin emergence and use. These papers include but not limited to the work by Michael Grieves back in 2002/2014/2017 in conceptualizing digital twin in product life cycle management, Glassen and Stargel’s paper in 2012 that better defines the general understanding, Ierapetritou’s review paper in 2020 that comprehensively describes the application of digital twin, etc. As a review article, these contributions should be mentioned to enrich the intro/literature review section, or at the very least, be mentioned in the supplemental material.
7. Case 2.1: Instead of twinning the patients, the specific case study measures glucose only, similar to existing biomedical devices. How is this case new or how can this so-called “twin” be beneficial to the industry would require further discussion. Also, with Figure 2 and related discussion, the reviewer is unsure if a control strategy is incorporated into the mechanism (i.e., if the glucose is high, did the case apply any actions to the patient?). In addition, the case seems focused on monitoring and denoising, instead of prediction as depicted in Figure 1.
8. Case 2.2: It is understood that the case reconstructs an in-silico human heart through MRI images and uses it to simulate ECG results. The application of such model looks nice, but the key question is if the developed image and ECG model can help with the patients. Are we able to use the 3D reconstruct to target cardiac diseases or guide treatment plans?
9. The rest of the cases focus more on the application of different algorithms in developing data-driven models and computer vision tools, and the critical point of “twining target” remains unclear. This can be tied back to comment 5 as the paper does not seem to have an overall structure for the digital twin of concern.
Author Response
Thank you for your feedback. We revised our manuscript according to the comments below and indicated with line numbers the respective changes. Note that these line numbers only match when selecting "Review -> simple markup" in Word. Please find below our point by point response.
- Abstract Line 8 and related descriptions: Here the authors define digital twins to replicate patients. This “definition” or definitive statement can only be applied to this specific manuscript. Instead of confusing readers with this broad statement, the authors should revise and offer the definition for their specific case.
Thank you for your comments and feedback. We replaced the statement with “In this review article, we consider digital twins as virtual counterparts of real human patients.” (line 11). In the introduction, we added a reference for this definition (line 34) [Venkatesh et al, npj Digit. Med., 2022].
- Introduction Line 31: Again, “healthcare” alone is a very general term, and it can have different meaning to readers. The authors can better define the overall scope of the term and the application area.
We changed the wording in line 32 to “In several fields of healthcare, such as precision medicine, clinical trials, and public health”. We aim to review the methodological challenges associated with digital twins across a broad spectrum of healthcare domains, therefore we did not limit the scope to one particular field. We clarified this in the manuscript by adding our explicit aims and motivation for this narrative review in the introduction (line 45-51 and 64-66).
- Introduction paragraph starting line 51 and the leading paragraph of Section 2: This reviewer is unsure if the case studies presented are “reviews” of other papers or specific demonstrations developed by the authors. This distinction is important, and the authors should make it clear early in the paper.
We did not develop these case studies ourselves, but we provide a review of the papers on these case studies. To remove the confusion, we revised the last paragraph of the introduction and added our explicit aims (line 64-71).
- Section 2 Line 67-68: What is the relevance between the case studies and the annotated overviews of papers in the supplemental material? Why is this supplemental material being mentioned here? Please provide further explanation and connections in the text.
In the article, we present several case studies discussing the data types and methodology used for the digital twin. We believe it’s of added value to integrate all this information into one searchable table, where the reader is referred to further literature regarding a specific case or method, as well as provide some additional materials if relevant for the case study methodology. We explained the purpose of the supplementary material in line 81-86.
- Before going into specific case studies, the authors are recommended to provide their understanding or definition of “digital twin”, applying specifically to this application area. In the current manuscript, such an understanding/scope does not seem to be clear. More importantly, each case seems to have a different meaning, making it hard to grasp an overall structure. If the authors intend to highlight different components in constructing a digital twin, then both the overall framework and the component should be highlighted, preferably in figures.
We added a paragraph regarding the definitions of digital twins in the literature. We adopt a specific definition for healthcare and explain that, even though this definition can be applied to different digital twin applications, the underlying data and methodology differ considerably, leading to very different operationalization of this definition. With this review, we want to go deeper into the data and methodology, aiming to make it easier to identify the challenges and opportunities in future studies. This clarification is added in the introduction (line 39-51).
- Supplemental material: The authors have highlighted this summary Excel file in a couple of instances, but the overall idea is unclear to the reviewer. It seems to cover digital twin topics in specific healthcare areas but omits important papers on digital twin emergence and use. These papers include but not limited to the work by Michael Grieves back in 2002/2014/2017 in conceptualizing digital twin in product life cycle management, Glassen and Stargel’s paper in 2012 that better defines the general understanding, Ierapetritou’s review paper in 2020 that comprehensively describes the application of digital twin, etc. As a review article, these contributions should be mentioned to enrich the intro/literature review section, or at the very least, be mentioned in the supplemental material.
Thank you for the references. We added them in the main text when discussing the definition of digital twins. The aim of the supplementary is to help the reader quickly finding relevant literature regarding digital twin methodology in a specific case, therefore we tried to limit to literature directly relevant to one of the cases in terms of methodology. We added text to explain this in the first paragraph of Section 2 (line 81-86).
- Case 2.1: Instead of twinning the patients, the specific case study measures glucose only, similar to existing biomedical devices. How is this case new or how can this so-called “twin” be beneficial to the industry would require further discussion. Also, with Figure 2 and related discussion, the reviewer is unsure if a control strategy is incorporated into the mechanism (i.e., if the glucose is high, did the case apply any actions to the patient?). In addition, the case seems focused on monitoring and denoising, instead of prediction as depicted in Figure 1.
To clarify how this methodology can serve as a digital twin, we added a sentence “Addressing these issues is essential to be usable as a ’near-future’ digital twin: for example, if glucose levels are predicted to be too high or too low in the near future, the system can generate preventive alerts, prompting the patient to take appropriate actions, such as adjusting insulin dosage or dietary choices.” (line 98-102). Also see our response below to comment 9.
- Case 2.2: It is understood that the case reconstructs an in-silico human heart through MRI images and uses it to simulate ECG results. The application of such model looks nice, but the key question is if the developed image and ECG model can help with the patients. Are we able to use the 3D reconstruct to target cardiac diseases or guide treatment plans?
See our response below to comment 9.
- The rest of the cases focus more on the application of different algorithms in developing data-driven models and computer vision tools, and the critical point of “twining target” remains unclear. This can be tied back to comment 5 as the paper does not seem to have an overall structure for the digital twin of concern.
We agree that the case study reviews focus on the methodology and data, without going into detail regarding implications of the digital twin of each case study on the respective healthcare domain. Generally, the digital twin should enable clinicians “gain valuable in-sights, optimize treatment strategies, and deliver personalized care” (line 41-42). We think that providing an in-depth study of the (potential) impact of each digital twin on healthcare, though extremely interesting, is outside the scope of this narrative methodological review.
Reviewer 2 Report
The authors focused in their manuscript on digital twins in health care. The basic idea reviewing the use of digital twins in health care and identifying challenges and opportunities is good. But the text needs additional work to be ready for publication.
The authors declare the manuscript as a review. I am afraid it does not satisfy the basic requirements laid on a proper review.
They present several case studies of use/potential use of digital twins in different medical areas. It is not clear why exactly these areas were selected since they are not interconnected.
For each area there are only few references. The description is rather superficial. No deeper analysis is present.
The total number of references is not satisfactory.
There is mentioned in the text that there is a spreadsheet in Supplementary Materials. However, the link in the manuscript is not working and thus the spreadsheet is not available.
English is good. Maybe some formulations can be improved.
Author Response
Thank you for your feedback. We revised our manuscript and indicated with line numbers the respective changes. Note that these line numbers only match when selecting "Review -> simple markup" in Word. Please find below our response.
The authors focused in their manuscript on digital twins in health care. The basic idea reviewing the use of digital twins in health care and identifying challenges and opportunities is good. But the text needs additional work to be ready for publication.
The authors declare the manuscript as a review. I am afraid it does not satisfy the basic requirements laid on a proper review.
They present several case studies of use/potential use of digital twins in different medical areas. It is not clear why exactly these areas were selected since they are not interconnected.
For each area there are only few references. The description is rather superficial. No deeper analysis is present.
The total number of references is not satisfactory.
There is mentioned in the text that there is a spreadsheet in Supplementary Materials. However, the link in the manuscript is not working and thus the spreadsheet is not available.
Thank you for your feedback. This manuscript serves as a narrative review, where our primary aim is to review the challenges and opportunities of the methodology and data types used to construct digital twins across several fields of healthcare. We intentionally selected case studies from different healthcare fields where digital twins are often constructed. They are selected as key showcases as they all aspire to construct digital twins, but differ considerably in terms of data and methodology used. The narrative review is not focused on the potentials and implications of the digital twin itself (in the respective context), but on the challenges and opportunities associated with the data and methods used to construct the twin.
To summarize, our aim is not to provide a systematic review, rather we believe that examining several case studies across different fields may provide a more in-depth study of the methods and data used, aiming to make it easier to identify the challenges and opportunities in future studies.
We revised the introduction in several places to better reflect our aims and scope, and explicitly mention two primary aims of our manuscript (line 64-71). We also added additional references regarding the definition of digital twins (line 39-42). We further added Table 1 to aid the reader in understanding the (main findings of the) case studies (page 11-13). Also, please find the spreadsheet (not yet hosted on the MDPI website) available as a supplementary file among the submitted documents. We also added an explanation of the role of the supplementary file (line 81-86).
Reviewer 3 Report
This paper is a review of the healthcare applications of digital twins. While the content is interesting, there are significant problems with its structure. To be blunt, it appears that the paper simply selected articles that appear to use digital twins without much coherence. With the current structure, it's difficult to understand what a digital twin actually is and what potential it holds. Therefore, I would like to request a significant restructuring.
First, I'd like the authors to provide a clear definition of digital twins in the introduction. Based on this, I suggest that each example mentioned in section 2 be summarized in a table. Suggested columns for this table are 1. Does this example meet the definition of a digital twin? 2. What data was entered to create the digital twin in this example? 3. What aspects of its physical counterpart can this digital twin replicate in the digital realm? 4. What possibilities does this offer?
With these changes, the similarities and differences of each example as a digital twin will become more apparent, thereby enhancing the reader's understanding.
I look forward to receiving the revised manuscript.
Author Response
Thank you for your feedback. We revised our manuscript according to the comments below and indicated with line numbers the respective changes. Note that these line numbers only match when selecting "Review -> simple markup" in Word. Please find below our response.
This paper is a review of the healthcare applications of digital twins. While the content is interesting, there are significant problems with its structure. To be blunt, it appears that the paper simply selected articles that appear to use digital twins without much coherence. With the current structure, it's difficult to understand what a digital twin actually is and what potential it holds. Therefore, I would like to request a significant restructuring.
First, I'd like the authors to provide a clear definition of digital twins in the introduction. Based on this, I suggest that each example mentioned in section 2 be summarized in a table. Suggested columns for this table are 1. Does this example meet the definition of a digital twin? 2. What data was entered to create the digital twin in this example? 3. What aspects of its physical counterpart can this digital twin replicate in the digital realm? 4. What possibilities does this offer?
With these changes, the similarities and differences of each example as a digital twin will become more apparent, thereby enhancing the reader's understanding.
I look forward to receiving the revised manuscript.
Thank you for your feedback and suggestions. This manuscript serves as a narrative review, where our primary aim is to review the challenges and opportunities of the methodology and data types used to construct digital twins across several fields of healthcare. We deliberately selected case studies from different healthcare fields where digital twins are often constructed. They are selected as key showcases as they all aspire to construct digital twins but differ considerably in terms of data and methodology used. We do not focus to provide a systematic review of the merits and implications of the digital twin itself, but on the challenges and opportunities associated with the data and methodology used. We revised the introduction in several places, in particular the last paragraph (line 64-71), to better reflect our aims and scope. We also added a paragraph regarding the definition of digital twins (line 39-42) followed by our motivation for this review.
We added a summary of our review in each case study in Table 1, describing the aim, input data, and methodology of the respective digital twin. We refer to it in the Discussion when summarizing our findings, page 11-13.
Round 2
Reviewer 2 Report
The manuscript has been significantly improved.
However, even a narrative review should pose a research question (or several research questions). The authors define aims of the manuscript. I would appreciate if they can formulate the corresponding research questions.
Section 2 Case studeis, lines 98 and 99: I am not convinced that the overview is comprehensive. It shows examples of digital twins in several healthcare domains. There are definitely other domain where digital twins were applied. The formulations must be precise and not misleading.
I would also expect deeper analysis in the Discussion part, in particular related to the type of data and methodologies used in individual presented case studies.
Language has been improved. Some sentences could be re-formulated, as for example those containing "overcoming the/these challenges", which is frequent in the Discussion section.
Author Response
The manuscript has been significantly improved.
However, even a narrative review should pose a research question (or several research questions). The authors define aims of the manuscript. I would appreciate if they can formulate the corresponding research questions.
[Note: The line numbers below are only correct when selecting “simple markup” under Review/track changes.]
Thank you for your comments. We formulated the three research questions that we aimed to address (listed below) and added it to the manuscript (p2 lines 69-73).
- Which data types and sources are important for the development of healthcare digital twins?
- What are prevailing methods and techniques employed in healthcare digital twin systems, and how do they vary in their applications?
- How can the challenges related to healthcare digital twin methods and data be transformed into opportunities?
Section 2 Case studeis, lines 98 and 99: I am not convinced that the overview is comprehensive. It shows examples of digital twins in several healthcare domains. There are definitely other domain where digital twins were applied. The formulations must be precise and not misleading.
We agree with the reviewer, indeed we did not intend to cover the whole application domain of digital twins. We removed “comprehensive” such that it reads as “the case studies reviewed below provide an overview of the methodological cornerstones of digital twins across different healthcare domains.” (p2 lines 88-89)
I would also expect deeper analysis in the Discussion part, in particular related to the type of data and methodologies used in individual presented case studies.
We extended the discussion to include a concise analysis of the methods and data used in the case studies (page 13-14: lines 445-448, 461-465, 480-491).
Language has been improved. Some sentences could be re-formulated, as for example those containing "overcoming the/these challenges", which is frequent in the Discussion section.
We have rewritten most of these instances to a more concise formulation (in particular: p14 lines 496-497). We further did a spelling and grammar check and corrected linguistic mistakes.
Reviewer 3 Report
Thank you for your revision. I think your manuscript has improved significantly.
Author Response
Thank you for your positive conclusion.